# Photo-degradation dynamics of five neonicotinoids: Bamboo vinegar as a synergistic agent for improved functional duration

**Rui Liang[1], Feng Tang[2], Jin Wang[2], Yongde Yue [2]***

**1** School of Resource & Environment, Anhui Agricultural University, Hefei, China, **2** State Forestry Administration Key Open Laboratory, International Centre for Bamboo and Rattan, Beijing, China

* yueyd@icbr.ac.cn

**Data Availability Statement:** All relevant data are within the manuscript and its Supporting Information files.

## Abstract

The effects of photo-degradation on the utilization of pesticides in agricultural production has been investigated. Various influencing factors were compared, with results showing that the initial pesticide concentration, light source, water quality and pH possessed different effects on neonicotinoids photo-degradation. The initial concentration and pH were found to be most critical effects. The photo-degradation rate decreased by a factor of 2–4 when the initial concentration increased from 5 mg L$^{-1}$ to 20 mg L$^{-1}$, particularly for acetamiprid and imidacloprid. The photo-degradation pathways and products of the five neonicotinoids were also investigated, with similar pathways found for each pesticide, except for acetamiprid. Degradation pathways mainly involved photo-oxidation processes, with products identified using liquid chromatography-quadrupole time-of-flight mass spectrometry (LC-Q-TOF-MS) found to be consistent with literature reported results. Bamboo vinegar exerted a photo-quenching effect on the neonicotinoids, with an improved efficiency at higher vinegar concentrations. The photo-quenching rates of thiamethoxam and dinotefuran were 381.58% and 310.62%, respectively, when a 30-fold dilution of vinegar was employed. The photo-degradation products in bamboo vinegar were identical to those observed in methanol, with acetic acid being the main factor influencing the observed quenching effects.

## Introduction

Neonicotinoids are a relevant class of insecticides, following pyrethroids, organophosphorus, and carbamates, successfully developed based on nicotine modifications [1,2]. Since entering the market at the end of the 1980s, neonicotinoids have been extensively employed in the control of piercing-sucking mouth insects including aphids, aleyrodidae and delphacidae, due to their good root uptake, broad spectrum, high efficiency and low toxicity to mammals [3,4]. Neonicotinoids function by selectively acting as effective agonists of the nicotine acetylcholine receptors (nAChRs) which play a key role in the nervous system of insects. These insecticides can consequently lead to nerve damage and eventually paralysis and death by causing severe

**Funding:** This research was funded by the financial supports from the National Science and Technology Infrastructure Program (2012BAD23B03).

**Competing interests:** The authors have declared that no competing interests exist.

neurotoxic effects. The discovery of new nicotine-related compounds can also be regarded as a milestone in insecticide research, potentially being able to provide an improved understanding of the functional properties of insect nAChRs [5,6]. However, the stability and functional duration of neonicotinoids is limited after conventional treatments due to their unique structures. A systematic study of the photo-degradation of neonicotinoids and the effect of different environmental conditions is therefore required to develop efficient synergistic agents to enhance insecticidal duration.

Most pesticides are sensitive to photolysis because of their structural properties; thus, photo-degradation commonly occurs for pesticides in the presence of light [7]. Photo-degradation of pesticides in water, a major degradation pathway after spraying, has a direct impact on the sustainability and degradation dynamics of pesticides [8,9]. This aqueous photo-degradation process can be influenced by a number of factors and may lead to two distinct phenomena, namely, photosensitization and photo-quenching [10,11]. Furthermore, photo-degradation itself is a complex process, as the photo-degradation pathways and formed intermediates/products of different (even the same) pesticides may greatly vary as a result of variable factors under different conditions [12,13]. Consequently, investigations on the effects of different factors on the photo-degradation of pesticides are of remarkable environmental significance. Presently, the photo-degradation of neonicotinoid insecticides in the environment has attracted significant research attention. However, most studies to date lack of a systematic approach.

Bamboo vinegar has recently attracted research interest as an environmental synergistic agent [14]. A brown liquid with unique smoky taste, bamboo vinegar is collected via condensation of gases during bamboo pyrolysis or distillation. The vinegar, which contains over 80 components, comprises 80–90% water and 7–11% organic acids, mostly acetic acid (approximately 6%), formic acid, butyric acid, propionic acid, phenols, ketones, aldehydes, alcohols, esters as well as other components [15]. Bamboo vinegar possesses antimicrobial and antioxidant activities, also promoting plant growth, being employed as a soil improver, fertilizer synergistic agent, feed supplement, edible fungi production supplement, etc. [16,17]. In addition, bamboo vinegar can be utilized as a pesticide synergistic agent in agricultural production activities [18]. However, studies on the utilization of bamboo vinegar as a light-stabilizing agent in pesticide formulations are scarce in literature, particularly for neonicotinoids.

In this study, the photo-degradation dynamics of imidacloprid, acetamiprid, clothianidin, thiamethoxam and dinotefuran (Fig 1) were investigated based on effects of initial concentration, light source, water quality and pH. Furthermore, the five neonicotinoids and the photo-degradation products were identified and characterized using liquid chromatography-quadrupole time-of-flight mass spectrometry (LC-Q-TOF-MS) (Figs A-J in S1 File), with degradation pathways also evaluated. The influence of bamboo vinegar on the photo-degradation of neonicotinoids was also determined, and the photo-degradation products in vinegar were investigated. Finally, the photo-degradation patterns of the five neonicotinoids were analyzed and their synergism with bamboo vinegar was evaluated to provide a scientific basis for an improved utilization of neonicotinoids.

## Materials and methods

### Chemicals

Standards of imidacloprid, acetamiprid, clothianidin, thiamethoxam and dinotefuran were purchased from Dr. Ehrenstorfer Industrial Corporation, Germany. Methanol, acetonitrile, and acetone were obtained from Xilong Chemical Reagent Co., China, and dichloromethane was purchased from Shanghai Zhenxing Chemical Reagent Co., China. Bamboo vinegar was provided by the International Centre of Bamboo and Rattan, Beijing, China.

**Fig 1. The structures of five neonicotinoids.**

### Neonicotinoids photo-degradation in water

All five neonicotinoids were dissolved in methanol as solvent to prepare standard solutions and analyzed using high-performance liquid chromatography (HPLC) with variable-wave-length ultraviolet detection to obtain standard curves. The initial concentration, light source, water quality, and pH were employed as parameters to systematically determine the photo-degradation characteristics of the neonicotinoids in water. The selected initial concentrations were 5 mg $L^{-1}$, 10 mg $L^{-1}$, and 20 mg $L^{-1}$; 20 mL of each solution was added to a stoppered quartz tube, and a high-pressure mercury lamp (300 W, 3300–5700 Lux, cold water reflux) was utilized as light source. The quartz tube was placed 7.0 cm away from the light source, and the temperature in the photo-degradation apparatus was carefully controlled (20 ± 2˚C). Each experiment was repeated three times.

A high-pressure mercury lamp and sunlight were employed to determine the influence of different light sources in the photo-degradation efficiency. Standard solutions of the

neonicotinoids were diluted to 5 mg L$^{-1}$ using ultra-pure water, and 20 mL of each solution was added to a stoppered quartz tube. Darkness experiments (in the absence of light) were conducted as control, and all samples were irradiated at different times.

Additionally, the standard solutions of the five neonicotinoids were diluted to 5 mg L$^{-1}$ using ultra-pure water, tap water or pond water, and sunlight was employed as light source to investigate the influence of water quality on neonicotinoid photo-degradation. Neonicotinoids were also diluted to 10 mg L$^{-1}$ using a disodium hydrogen phosphate-citric acid buffer to obtain pH values of 4.0, 5.5, 7.5, 8.5, and 10.0. A high-pressure mercury lamp was used for illumination to determine the pH effect on neonicotinoid photo-degradation.

All quantitative photo-degradation analyses were performed in an Agilent 1200 HPLC equipped with a Thermo Acclaim 120 C$_{18}$ column (4.6 mm × 250 mm, 5 μm). Mobile phase A was acetonitrile and mobile phase B was ultra-pure water, employing a 30:70 (A:B) solvent ratio. The flow rate, column temperature and injection volume were 1000 μL min$^{-1}$, 30°C, and 20 μL, respectively.

## Analysis of photo-degradation products

Standards of imidacloprid, acetamiprid, clothianidin, thiamethoxam and dinotefuran were prepared at a concentration of 10 mg L$^{-1}$ in methanol. Solutions were photo-treated under UV irradiation, and a darkness control was established. Samples were then filtered using a membrane filter with a pore size of 0.22 μm and analyzed using LC-Q-TOF-MS equipped with diode array detection.

LC separations were carried out in an Agilent 6540 Infinity II series LC system (Agilent Technologies, Santa Clara, CA) equipped with an Agilent Eclipse Plus C$_{18}$ column (2.1 mm × 150 mm, 1.8 μm). Mobile phase A was acetonitrile, mobile phase B was ultra-pure water; the elution gradient was as follows: 0–4 min, 25% A; 4–7 min, 25–80% A; and 7–10 min, 80–25% A. The flow rate, column temperature and injection volume were 250 μL min$^{-1}$, 25°C and 2 μL, respectively. An Agilent 6540 Q-TOF-MS (Agilent Technologies, Santa Clara, CA) equipped with an automatic jet stream electrospray ionization (AJS-ESI) interface was employed for assay development. The typical ion source parameters were as follows: capillary voltage, +4000/-3500 V; nozzle voltage, +500/-500 V; gas temperature, 300°C; gas flow rate, 5 L min$^{-1}$; sheath gas temperature, 250°C; sheath gas flow rate, 11 L min$^{-1}$; and nebulizer pressure, 45 psi.

## Influence of bamboo vinegar

All five neonicotinoids were dissolved in methanol (concentration 1000 mg L$^{-1}$) and combined with bamboo vinegar for a final neonicotinoid concentration of 10 mg L$^{-1}$ and bamboo vinegar dilution rates of 30- and 100-fold. The mixed solutions (20 mL) were placed in stoppered quartz tubes and set in the photo-degradation apparatus 7 cm away from the high-pressure mercury lamp. The temperature was carefully controlled (20 ± 2°C) in the photo-degradation apparatus. Each experiment was conducted in triplicate, and samples were analyzed after different times of irradiation.

## Analytical methodologies

The photo-degradation efficiency was calculated using the following formula:

$$Photodegradation\ rate(\%) = \frac{C_d - C_l}{C_d} \times 100,$$

where $C_d$ and $C_l$ are neonicotinoid concentrations (darkness control) and different experiments, respectively.

The photo-degradation kinetics of the five neonicotinoids in solution were described by a first-order kinetic equation:

$$C_t = C_0 \times e^{-kt},$$

where $C_0$ and $C_t$ are the initial concentration of the neonicotinoids and that at an arbitrary time t, respectively, and $k$ is the photo-degradation rate constant.

## Results and discussion

### Effect of the neonicotinoid initial concentration on photo-degradation

Fig 2 illustrates the rate constant dynamics and attenuation curve for the photo-degradation of the five neonicotinoids at different initial concentrations. The rate constants of the aqueous neonicotinoids were negatively correlated with the initial concentration (5 mg L$^{-1}$, 10 mg L$^{-1}$, and 20 mg L$^{-1}$). The half-lives of imidacloprid, acetamiprid, clothianidin, thiamethoxam and dinotefuran were 45.2 min, 902.1 min, 34.7 min, 21.2 min and 163.2 min, respectively, at an initial concentration of 20 mg L$^{-1}$. Comparably, 17.6 min, 223.6 min, 14.7 min, 10.6 min and

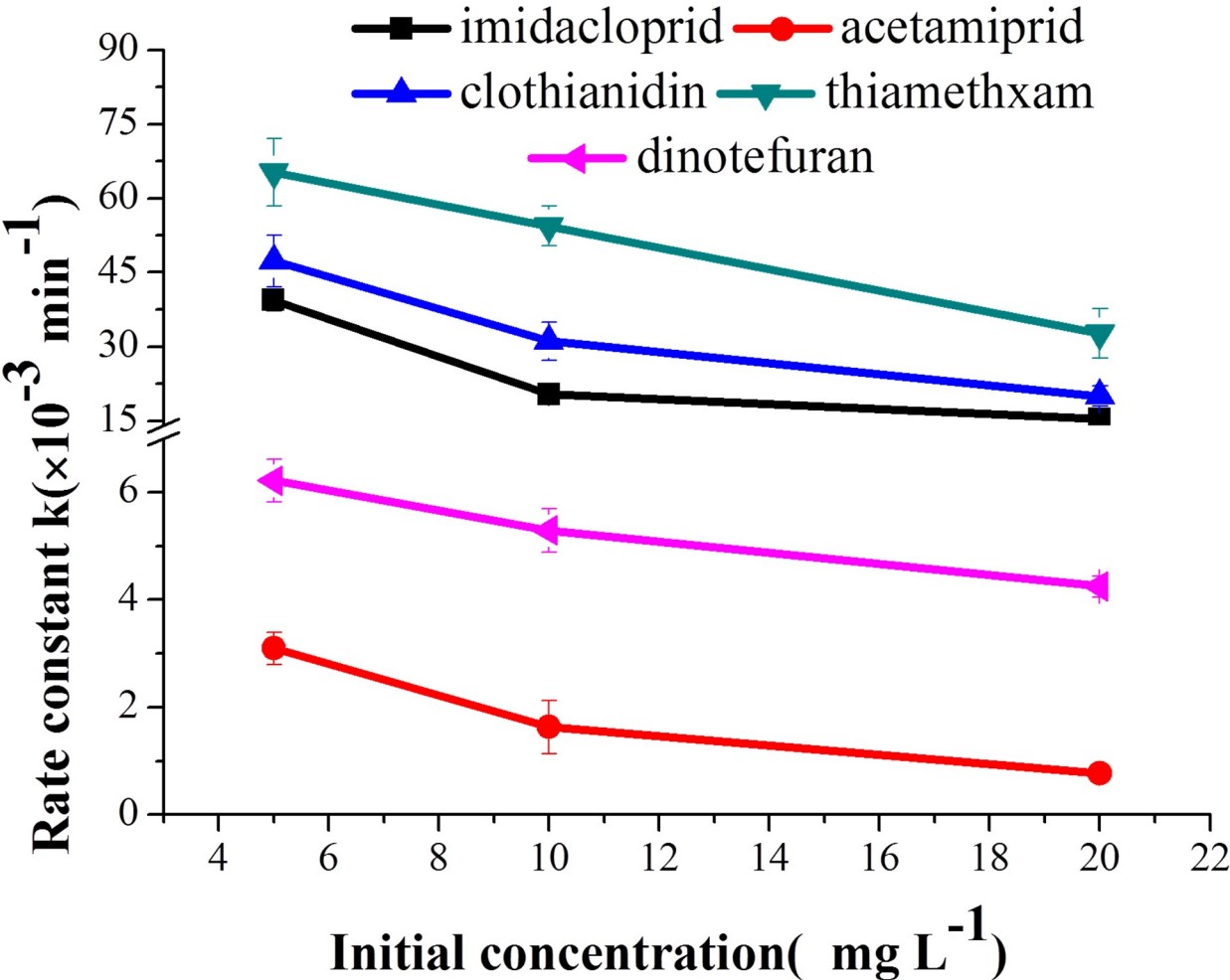

**Fig 2. Photo-degradation of five neonicotinoids at different initial concentrations.** a = imidacloprid, b = acetamiprid, c = clothianidin, d = thiamethoxam, and e = dinotefuran.

111.4 min, respectively, were obtained as half-lives for the same insecticides at an initial concentration of 5 mg L$^{-1}$. Larger initial pesticide concentrations correspond to a higher content of pesticide molecules in aqueous solution; this ensures more photon absorption competition and thus less light energy absorbed per unit molecule, which leads to a lower photo-degradation rate [19,20]. In addition, many intermediate products can form from neonicotinoids photo-degradation and then compete for photons, thereby reducing the quantum yield of absorbed photons [21–23].

## Effect of the light source

In general, pesticides are exposed to sunlight or other light sources to investigate the effect of the light source on their photo-degradation behaviour [24,25]. Sunlight and a high-pressure mercury lamp were herein employed to evaluate such effects on the photo-degradation of the five neonicotinoids in this study, with results shown in Fig 3. Photo-degradation occurred under irradiation with both light sources, following a first-order kinetic equation. The photo-degradation rate under high-pressure mercury lamp irradiation was significantly higher as compared to that observed under sunlight irradiation. According to the regression equation of the photo-degradation reaction kinetics, the half-life of thiamethoxam under sunlight (20.8 min) was comparably higher to that under high-pressure mercury lamp irradiation (10.8 min) by a factor of 2, whereas the half-lives of the other pesticides were higher by a factor of over 1.5.

This difference may be related to the emission spectra of the light sources and the absorption spectra of the neonicotinoids. The emission spectrum of a high-pressure mercury lamp ranges from 190 nm to 1000 nm [26], while the maximum absorption wavelengths of the tested neonicotinoids are concentrated in the ultraviolet region near 260 nm [27]. Therefore, under irradiation from the high-pressure mercury lamp, photo-degradation of the neonicotinoid pesticides can rapidly occur. In contrast, the emission spectrum of sunlight is mainly in the visible region and less in the ultraviolet region; thus, the neonicotinoids absorb only part of sunlight. The absorption efficiency of photon energy under sunlight irradiation is comparatively lower to that of the high-pressure mercury lamp, which results in the expected lower photo-degradation rate under sunlight irradiation [28–30].

## Effect of water quality

The effects of ultra-pure water, tap water and pond water on the photo-degradation of neonicotinoids were subsequently investigated, with results depicted in Fig 4.

No significant differences could be observed in the photo-degradation rates or half-lives of thiamethoxam or dinotefuran in the different water media. Imidacloprid exhibited the largest differences, with half-lives of 19.5 min, 67.4 min, and 73.9 min in ultra-pure water, tap water and pond water, respectively. The photo-degradation rate constants of acetamiprid were 0.00390 K min$^{-1}$, 0.00385 K min$^{-1}$ and 0.00288 K min$^{-1}$ in ultra-pure, tap, and pond water, respectively, whereas those of clothianidin were 0.01350 K min$^{-1}$, 0.01190 K min$^{-1}$, and 0.01090 K min$^{-1}$, respectively.

Ultra-pure water exhibits the lowest conductivity ($1.0 \times 10^{-4}$ S m$^{-1}$), lowest amount of dissolved substances and almost no light absorption and transmission scattering; thus, pesticides exhibited the fastest photo-degradation rate in this medium [31,32]. The electrical conductivity of tap water ($1.4 \times 10^{-2}$ S m$^{-1}$) is higher with respect to that of ultra-pure water due to the presence of certain ions, soluble substances, and so forth. These substances can have a shielding effect in the absorption and transmission of light and may thus play a role in quenching the photo-degradation of pesticides in tap water [33,34]. Pond water contains a large amount of inorganic ions and dissolved organic matter, including humic acids, which can produce singlet

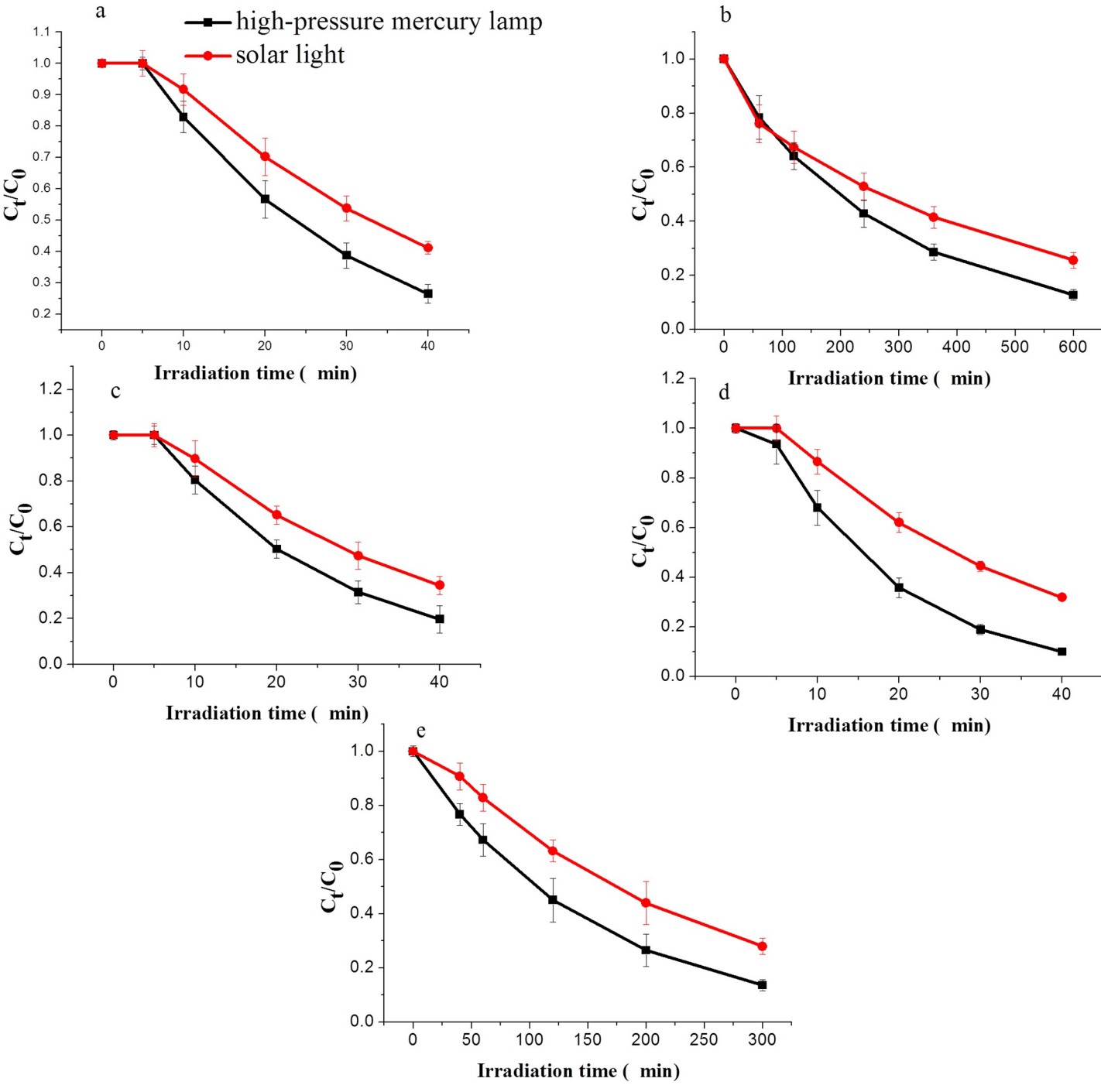

**Fig 3. Photo-degradation of five neonicotinoids under different light source irradiation.** a = imidacloprid, b = acetamiprid, c = clothianidin, d = thiamethoxam, and e = dinotefuran.

oxygen molecules with a stronger oxidation capacity as compared to molecular oxygen when exposed to sunlight [35,36]. These molecules can rapidly react with organic pollutants in water, thus promoting pollutant decomposition. However, humic acids and other organic matter may also reduce the photo-degradation rate of pesticides through competitive absorption of photons [37–39]. Importantly, dichromate oxidizability ($COD_{Cr}$) of pond water (126.5 mg

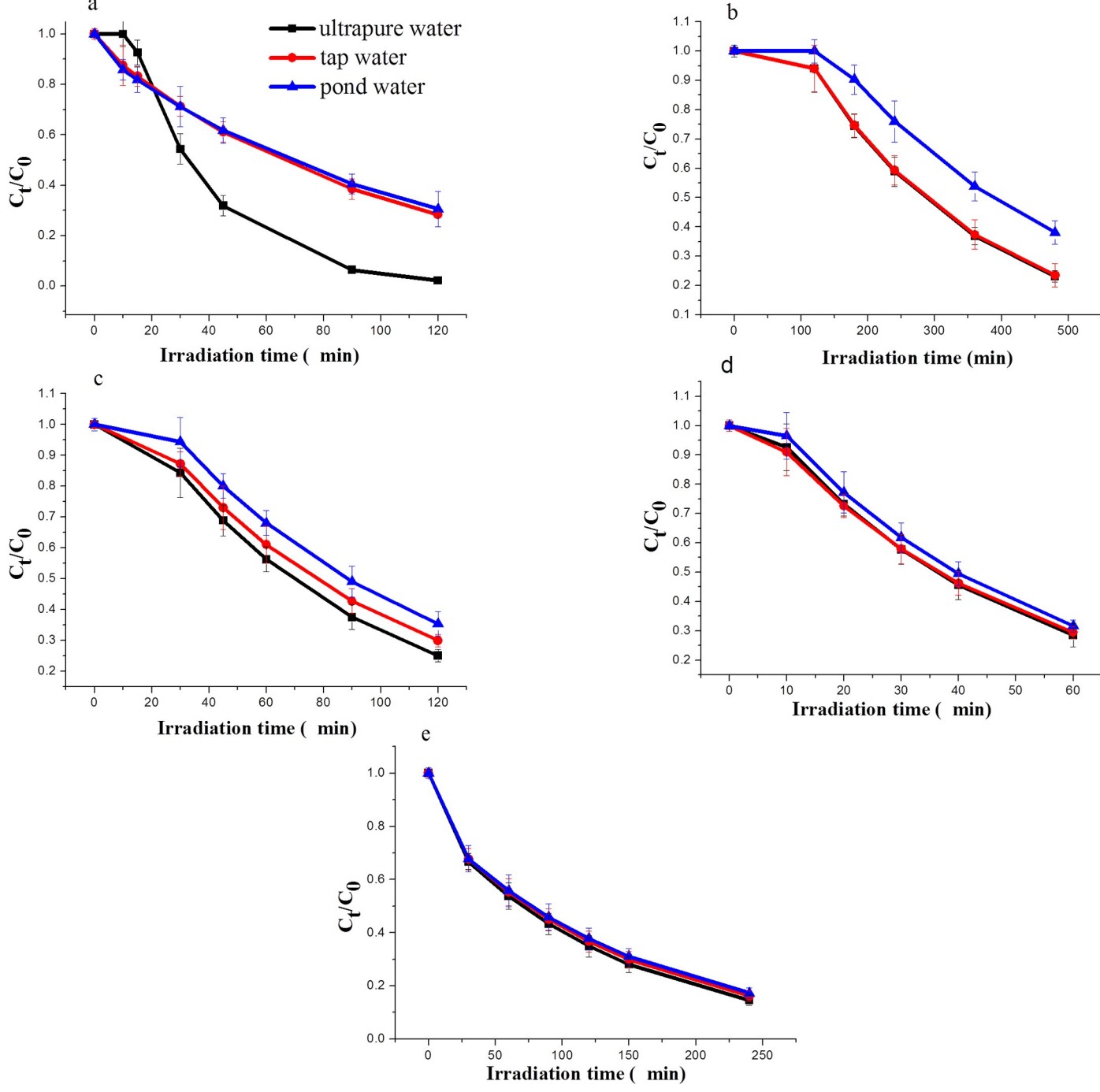

**Fig 4. Photo-degradation of five neonicotinoids in water of different quality.** a = imidacloprid, b = acetamiprid, c = clothianidin, d = thiamethoxam, and e = dinotefuran.

L$^{-1}$) was found to be significantly higher with respect to ultra-pure water and tap water (0 mg L$^{-1}$ and 1.7 mg L$^{-1}$, respectively). In view of these premises, water quality seem to influence photo-degradation rates via competitive absorption of photons as dominant process [40–42].

The photo-degradation kinetics of pesticides in natural water are related to the physical and chemical properties of water as well as the absorption and transmission of light by soluble

substances in the water and the differences in the photochemical reactions [43,44]. Tap water and pond water contain a large amount of ions including $NO_3^-$, $SO_4^{2-}$, and $Cl^-$, and soluble organic matter, which exhibit different behaviours in terms of photosensitization and photo-quenching of different pesticides [45,46]. No significant differences could be observed in the photo-degradation rates of imidacloprid and clothianidin in tap water and pond water, acetamiprid in ultra-pure water and tap water or thiamethoxam and dinotefuran in all three types of employed water.

## Effect of pH

The pH has a significant influence on the photo-degradation of organic pollutants [47,48]. In this study, a disodium hydrogen phosphate-citric acid buffer was employed to obtain pH values of 4.0, 5.5, 7.5, 8.5, and 10.0 in order to investigate pH effects on the photo-degradation of neonicotinoids. As listed in Table 1, the photo-degradation rate was low in an acidic buffer solution, whereas photodegradation rates were relatively high (and generally increased at increasing pH) in neutral and alkaline buffer solutions. At pH 4.0, the photo-degradation rates of all neonicotinoids were the lowest, with that of imidacloprid being similar at all tested pH values. Furthermore, the degradation rate of acetamiprid increased at increasing pHs, whereas that of clothianidin increased at pH 8.5 and 10.0. The photo-degradation rates of thiamethoxam and dinotefuran were similar at pH 7.5, 8.5, and 10.0.

Neonicotinoids readily produce hydroxyl radicals in alkaline solutions as compared to in acidic and neutral solutions, with such radicals being correlated to greater pesticide degradation [49,50]. The acidic buffer solution is more likely to absorb photons near 260 nm, maximum absorption wavelength of neonicotinoids, consequently competing with photon absorption (for photo-degradation) of the five pesticides, and resulting in reduced photo-degradation rates [51,52]. In alkaline solution, an increase in photo-degradation rates can be correlated to a higher photo-quantum yield and protonation level of the excited pesticide molecules [53–55].

## Photo-degradation products of neonicotinoids

The photo-degradation products of the five neonicotinoids in methanol were subsequently investigated. The product peaks in the total ion flow (TIC) diagram were compared to those in the parent standard TIC diagram. The mass spectra of the products was collected, and the attributes of the characteristic ions were analysed. The products and possible photo-degradation pathways were then compared to those reported in the existing literature.

As shown in Fig 5A, the photo-degradation product of imidacloprid is $C_7H_{14}N_2O_2$, in good agreement to that previously reported by Ding et al. [56]. The possible photo-degradation pathway proceeds via an imidacloprid molecule absorbing a photon under illumination, followed by N-N bond cleavage and $NO_2$ removal from the molecule. The C = N bond then combines with a hydroxyl radical (·OH) in water to form a C = O bond via substitution reaction [57].

The possible photo-degradation pathway of acetamiprid is shown in Fig 5B, for which $C_{10}H_{12}N_4O$ was detected as photo-degradation product. Such pathway for acetamiprid can be stimulated under light irradiation. A Cl atom linked to the pyridine ring is replaced by O, forming a double bond, with C bonds at positions 1 and 4 on the pyridine ring leading to bicyclic olefins [58, 59].

Fig 5C illustrates the possible photo-degradation pathway of clothianidin ($C_6H_8N_3OSCl$ as photo-degradation product), similar to that observed for imidacloprid, and different from results reported by Mulligan et al. [60].

**Table 1. Photo-degradation of five neonicotinoids at different pH values.**

| Neonicotinoids | pH value | Kinetic equation $\ln C_t = \ln C_0 - kt$ | Rate constant K $min^{-1}$ | Half-life period $T_{1/2}$ $min^{-1}$ |
|---|---|---|---|---|
| Imidacloprid | 4.0 | $\ln C_t = 2.1351 - 0.0121t$ | 0.01210 | 57.3a |
| | 5.5 | $\ln C_t = 2.1757 - 0.0198t$ | 0.01980 | 35.0b |
| | 7.5 | $\ln C_t = 2.1412 - 0.0208t$ | 0.02080 | 33.3b |
| | 8.5 | $\ln C_t = 2.1504 - 0.0213t$ | 0.02130 | 32.5b |
| | 10.0 | $\ln C_t = 2.1609 - 0.0220t$ | 0.02200 | 31.5b |
| Acetamiprid | 4.0 | $\ln C_t = 2.0396 - 0.0704t$ | 0.00117 | 590.8a |
| | 5.5 | $\ln C_t = 2.1135 - 0.0768t$ | 0.00128 | 541.5b |
| | 7.5 | $\ln C_t = 2.1521 - 0.0909t$ | 0.00152 | 457.5c |
| | 8.5 | $\ln C_t = 2.2300 - 0.1075t$ | 0.00179 | 386.9d |
| | 10.0 | $\ln C_t = 2.1022 - 0.1305t$ | 0.00218 | 318.7e |
| Clothianidin | 4.0 | $\ln C_t = 1.9161 - 0.0210t$ | 0.02100 | 33.0a |
| | 5.5 | $\ln C_t = 1.8984 - 0.0191t$ | 0.01910 | 36.3a |
| | 7.5 | $\ln C_t = 1.8803 - 0.0225t$ | 0.02250 | 30.8a |
| | 8.5 | $\ln C_t = 1.8988 - 0.0256t$ | 0.02560 | 27.1b |
| | 10.0 | $\ln C_t = 1.9102 - 0.0259t$ | 0.02590 | 26.8b |
| Thiamethoxam | 4.0 | $\ln C_t = 1.8123 - 0.0250t$ | 0.02500 | 27.7a |
| | 5.5 | $\ln C_t = 1.9628 - 0.0425t$ | 0.04250 | 19.3b |
| | 7.5 | $\ln C_t = 2.1090 - 0.0561t$ | 0.05610 | 12.4c |
| | 8.5 | $\ln C_t = 2.1701 - 0.0605t$ | 0.06050 | 11.5c |
| | 10.0 | $\ln C_t = 2.1357 - 0.0571t$ | 0.05710 | 12.1c |
| Dinotefuran | 4.0 | $\ln C_t = 1.9168 - 0.0120t$ | 0.01200 | 57.8a |
| | 5.5 | $\ln C_t = 2.0342 - 0.0146x$ | 0.01460 | 47.5b |
| | 7.5 | $\ln C_t = 2.3123 - 0.0211t$ | 0.02110 | 32.9c |
| | 8.5 | $\ln C_t = 2.4425 - 0.0239t$ | 0.02390 | 29.0c |
| | 10.0 | $\ln C_t = 2.2540 - 0.0224t$ | 0.02240 | 30.9c |

As shown in Fig 5D, the photo-degradation product of thiamethoxam is $C_6H_8SN_3OCl$, with a plausible photo-degradation pathway involving the absorption of a photon by thiamethoxam molecules which induces N-N bond cleavage followed by $-NO_2$ removal. As indicate before, the C = N bond combines with a hydroxyl radical (·OH) in water and forms a C = O bond through a substitution reaction; the five-membered ring is finally separated $-C_2H_2O$ to obtain the product [61–63].

The possible photo-degradation pathway of dinotefuran is illustrated in Fig 5E, leading to the formation of $C_7H_{14}N_2O_2$, identical to that observed for imidacloprid and clothianidin [64–66].

On the basis of the identified photo-degradation products of the five neonicotinoids in methanol, it can be inferred that the degradation pathways mainly consist of photo-oxidation processes [67,68].

## Effect of bamboo vinegar

The effect of bamboo vinegar on the photo-degradation of neonicotinoids was also investigated, with results illustrated in Fig 6. The addition of bamboo vinegar significantly decreased the photo-degradation rate of the five neonicotinoids. Under the irradiation of the high-pressure mercury lamp, bamboo vinegar diluted 30-fold and 100-fold had significant effects on the photo-degradation half-lives of the pesticides, where the photo-degradation rate decreased at increasing vinegar concentration.

**Fig 5. Photo-degradation pathways and products of five neonicotinoids.** a = imidacloprid, b = acetamiprid, c = clothianidin, d = thiamethoxam, and e = dinotefuran.

The photo-quenching rates of imidacloprid, acetamiprid, clothianidin, thiamethoxam, and dinotefuran reached 67.35%, 125.00%, 143.13%, 381.58%, and 310.62%, respectively, when the neonicotinoids were mixed with diluted 30-fold bamboo vinegar. Comparably, 6.49%, 28.57%, 34.78%, 114.02%, and 129.61%, respectively, accounted for photo-quenching rates of the same insecticides mixed with diluted 100-fold vinegar. These findings indicate that bamboo vinegar has a remarkable photo-stabilizing effect on the five neonicotinoids under high-pressure mercury lamp irradiation. The dependency of the photo-quenching effect of bamboo vinegar on the degree of dilution was similar for all five samples; as bamboo vinegar concentration increased, its light stabilization of the neonicotinoids became more significant. In addition, 30-fold diluted bamboo vinegar exhibited the most significant effect on the photo-stability of thiamethoxam and dinotefuran, for which the photo-quenching rates increased by a factor over 3. All photo-degradation products of the five neonicotinoids in bamboo vinegar were identical to those observed in methanol.

pH values of diluted 30-fold and 100-fold bamboo vinegar were 3.40 and 3.72, respectively, whereas that of ultra-pure water was 6.50. According to the above experimental results, the photo-degradation rate of the five neonicotinoids decreases at increasing acidity (from pH 3.40 to 6.50) in good agreement with previous reports of pH being reported as an influencing factor [69,70]. Since bamboo vinegar is a reddish-brown solution, natural pigments may compete with neonicotinoids for photon absorption [71,72]. With an increase in concentration of bamboo vinegar, the concentration of pigments also increases, inducing significant photon competition [73–75]. This competition is consistent with the observed correlation between the dilution degree of bamboo vinegar and its light stabilization effect. The potential main components involved in the quenching of neonicotinoid photo-degradation in bamboo vinegar include acetic acid and pigments as previously reported [76–79].

## Conclusions

The photo-degradation dynamics of five neonicotinoids and the influence of the initial concentration, light source, water quality and pH were investigated in this work. The photo-degradation half-lives of the neonicotinoids increased by nearly 1-fold when the initial neonicotinoid concentration decreased from 20 to 5 mg L$^{-1}$ (except for that of acetamiprid which increased by a factor of 3). The photo-degradation rates of the five neonicotinoids under high-pressure mercury lamp irradiation were comparably higher (by nearly a factor of 1.5) to those under sunlight irradiation. Photo-degradation rates were also different depending on the type of employed aqueous media, being the highest in ultra-pure water, followed by tap water and finally pond water, following a similar trend in terms of pH (lowest in acidic buffer solution and relatively high in alkaline and neutral buffer solutions).

Photo-degradation pathways of imidacloprid, clothianidin, thiamethoxam and dinotefuran were similar, mainly consisting of photo-oxidation processes. Bamboo vinegar exhibited a quenching effect on the photo-degradation of the five neonicotinoids, with this effect became significant at high vinegar concentrations (i.e. photo-quenching rates of thiamethoxam and dinotefuran were 381.58% and 310.62%, respectively, for 30-fold diluted vinegar). The photo-degradation rates of neonicotinoids decrease with increasing acidity, with acetic acid in bamboo vinegar considered as main factor influencing the observed photo-quenching effect.

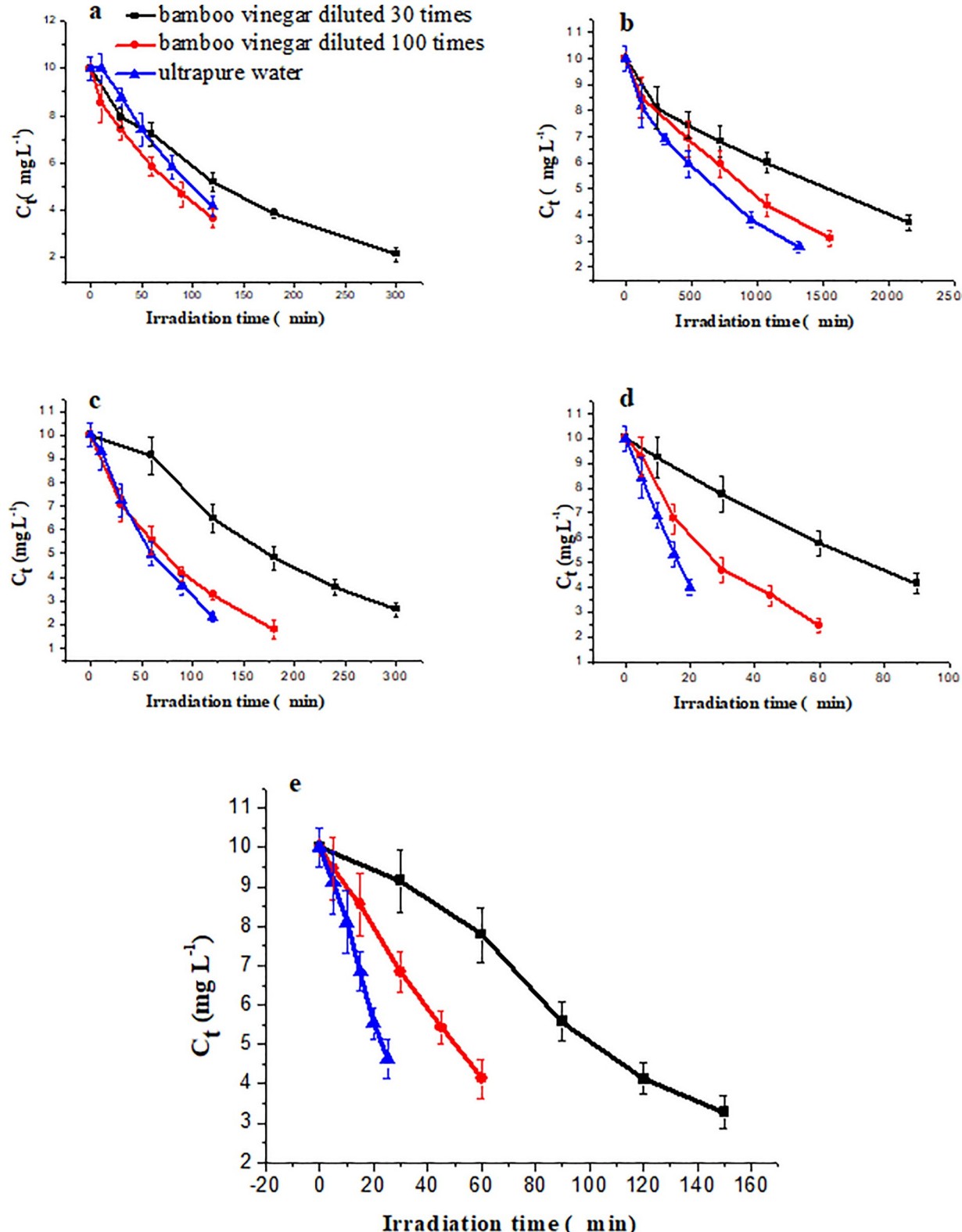

**Fig 6. The effect of bamboo vinegar on the photo-degradation of five neonicotinoids.** a = imidacloprid, b = acetamiprid, c = clothianidin, d = thiamethoxam, and e = dinotefuran.

These results have great significance in terms of improving functional lifetimes of neonicotinoids and the environmental friendliness of pesticides via addition and concentration adjustment of bamboo vinegar.

## Supporting information

**S1 File.** Figs A-J in Supporting Information file summary of the results of extraction ionic and ion fragmentation of neonicotinoids and photo-degradation products.
(DOCX)

## Author Contributions

**Data curation:** Rui Liang.

**Formal analysis:** Rui Liang.

**Investigation:** Rui Liang.

**Methodology:** Rui Liang, Jin Wang.

**Supervision:** Feng Tang, Yongde Yue.

**Writing – original draft:** Rui Liang.

**Writing – review & editing:** Rui Liang, Jin Wang.

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
