## [Decision Letter · Decision Letter 0]

12 Aug 2019

PONE-D-19-17188

Photo-degradation dynamics of five neonicotinoids: bamboo vinegar as synergistic agent for improved duration

PLOS ONE

Dear Dr. Yue,

Thank you for submitting your manuscript to PLOS ONE. After careful consideration, we feel that it has merit but does not fully meet PLOS ONE’s publication criteria as it currently stands. Therefore, we invite you to submit a revised version of the manuscript that addresses the points raised during the review process.

We would appreciate receiving your revised manuscript by Sep 26 2019 11:59PM. To enhance the reproducibility of your results, we recommend that if applicable you deposit your laboratory protocols in protocols.io, where a protocol can be assigned its own identifier (DOI) such that it can be cited independently in the future. For instructions see: http://journals.plos.org/plosone/s/submission-guidelines#loc-laboratory-protocols

We look forward to receiving your revised manuscript.

Kind regards,

Yulin Gao

Academic Editor

PLOS ONE

Journal Requirements:

Review Comments to the Author

Reviewer #1: Overall, this is a systematical study on investigating the photo-degradation of several neonicotinoids under various influencing factors including the initial concentration, light source, water quality and pH value. Moreover, bamboo vinegar has been found that it can serve as an efficient synergistic agent to improve the photo-stability of neonicotinoids, which is the major novelty of this manuscript. The organization and presentation of the manuscript are good. In general, this manuscript is suitable for publication in PLOS ONE after some major revisions.

1. The typical chromatograms of the photo process including photo-degradation intermediates and starting material of the neonicotinoid should be provided.

2. The possible photo-degradation products should be identified and confirmed by comparison with the authentic standards. The possible photo-degradation pathway and product of acetamiprid should be carefully reconsidered.

3. Lines 269-270, by separating and identifying the photo-degradation products of the five neonicotinoids in methanol, it can be inferred that the pathways are mainly photo-oxidation processes. Have you separated and identified these photo-degradation products?

4. Lines 301-302, it can be observed that the main factors that quench the photo-degradation of neonicotinoids in bamboo vinegar are possibly acetic acid and pigments. To confirm your hypothesis, you’d better to use acetic acid and one pigment to test the quenching effect.

Reviewer #2: Nice work, certainly worth publication in PLOS One after some minor revisions. The topic of the manuscript is of high importance in the field and results are very significant, will be useful to scientists working in photodegradation of organic contaminants. The manuscript is well written (perhaps some thorough reading needed to polish and correct typos and mistakes in the final version) and all results are of high interest, providing relevant insights on photodegradation intermediates and improvement use of pesticides by the synergistic effect with bamboo vinegar. The authors are kindly requested to shorten the conclusions (currently too long) and maybe redo some of the Figures of the photodegradation intermediates to ensure high quality in the final manuscript. Otherwise, a nice work worth publication in PLOS One.

Reviewer #3:  Your manuscript submission is interesting and contains information that our readers would find useful.  Unfortunately, the quality of the language is not up to the standards of the journal.

---

## [Author Response · Author response to Decision Letter 0]

23 Aug 2019

Dear Dr Yulin Gao, 

We have revised the manuscript in response to the editor and reviewers’ advice, and the following shows the point by point response to the reviewers’ comments. Additionally, we made some other revisions. All the changes in separate file and labeled ‘Revised Manuscript with Track Changes’ were highlighted in RED.

The language of this manuscript was carefully revised by ‘Elsevier Language Editing’.

We appreciate very much for your time spent on our manuscript and the referees for their valuable suggestions and comments. The revised manuscript has been resubmitted to your journal. We are looking forward to hearing from your final decision when it is made. 

Best wishes

Sincerely, 

Yongde Yue

22-Aug-2019 

Response to reviewers:

Reviewer: 1 

1. The typical chromatograms of the photo process including photo-degradation intermediates and starting material of the neonicotinoid should be provided. 

Response: We have noticed this problem, and agree with you. The necessary information about typical chromatograms of the photo process has been submitted as ‘Supporting Information’ file.

2. The possible photo-degradation products should be identified and confirmed by comparison with the authentic standards. The possible photo-degradation pathway and product of acetamiprid should be carefully reconsidered. 

Response: We have noticed this problem. The photo-degradation product of acetamiprid was identified using LC-Q-TOF-MS with a diode array detector, and which was compared with the results of PhD dissertation ‘Study on Hydrolytic and Photolytic Behavior of Acetamiprid in Environment’ that was completed by Dr. Guohong Xie. 

3. Lines 269-270, by separating and identifying the photo-degradation products of the five neonicotinoids in methanol, it can be inferred that the pathways are mainly photo-oxidation processes. Have you separated and identified these photo-degradation products? 

Response: We have modified this sentence. We didn’t separated the photo-degradation products, and standards of five neonicotinoids were prepared at 10 mg/L using methanol, then were photo-treated using the high-pressure mercury lamp, and a darkness control was set. Finally, the photo-degradation product samples were detected using LC-Q-TOF-MS. 

4.Lines 301-302, it can be observed that the main factors that quench the photo-degradation of neonicotinoids in bamboo vinegar are possibly acetic acid and pigments. To confirm your hypothesis, you’d better to use acetic acid and one pigment to test the quenching effect.

Response: We have noticed this statement, and agree with you. We have modified this sentence. In our study about the photo-degradation of five neonicotinoids with different pH value, in an acidic buffer solution, the photo-degradation rate was low, whereas in neutral and alkaline buffer solutions, the rate was relatively high and generally increased with increasing pH. And then compared with literature which listed as 47, 48, and 69-79 in the manuscript, the hypothesis was obtained. Of course, we will confirm that in the following in-depth study.

Thanks to reviewer’s suggestion, it’s very carefully.

Reviewer: 2 

Thank you for such high rating of this manuscript. We have polished and corrected typos and mistakes, and the language of this manuscript was carefully revised by ‘Elsevier Language Editing’. We have modified the conclusions part according the suggestion.

Reviewer: 3 

Thank you for such high rating to the results of this manuscript, and we are regret to the quality of the language. Then, we have polished and corrected typos and mistakes, and the language of this manuscript was carefully revised by ‘Elsevier Language Editing’. 

Thanks to reviewer’s comments and suggestions.

---

## [Decision Letter · Decision Letter 1]

4 Sep 2019

[EXSCINDED]

PONE-D-19-17188R1

Photo-degradation dynamics of five neonicotinoids: bamboo vinegar as synergistic agent for improved duration

PLOS ONE

Dear Dr. Yue,

Thank you for submitting your manuscript to PLOS ONE. After careful consideration, we feel that it has merit but does not fully meet PLOS ONE’s publication criteria as it currently stands. Therefore, we invite you to submit a revised version of the manuscript that addresses the points raised during the review process.

We would appreciate receiving your revised manuscript by Oct 19 2019 11:59PM. To enhance the reproducibility of your results, we recommend that if applicable you deposit your laboratory protocols in protocols.io, where a protocol can be assigned its own identifier (DOI) such that it can be cited independently in the future. For instructions see: http://journals.plos.org/plosone/s/submission-guidelines#loc-laboratory-protocols

We look forward to receiving your revised manuscript.

Kind regards,

Yulin Gao

Academic Editor

PLOS ONE

 Review Comments to the Author

Reviewer #3: Your manuscript submission is interesting and contains information that our readers would find useful. AND also this revision looks more better than before, Unfortunately, the quality of the language is not up to the standards of the journal.  language error still can be see in the current version.I suggest you may wish to consider having your paper professionally edited for English language by a native English speaker and/or a professional language editing service here before resubmitting this manuscript： http://www.zn16289208.icoc.bz/ http://www.unlecture.com/http://www.goodenglish.com/http://www.journalexpert.com/which specialise in editing scientific and technical research papers written in English by authors who are not native English speakers.Please note that while this service will greatly improve the readability of your paper, it does not guarantee acceptance of your paper by the journal. 

7. PLOS authors have the option to publish the peer review history of their article (what does this mean?). If published, this will include your full peer review and any attached files.

Reviewer #1: No

Reviewer #2: Yes: Rafael Luque

---

## [Author Response · Author response to Decision Letter 1]

14 Sep 2019

Dear Dr Yulin Gao, 

We have revised the manuscript in response to the editor and reviewers’ advice, and the following shows the point by point response to the reviewers’ comments. Additionally, we made some other revisions. All the changes in separate file and labeled ‘Revised Manuscript with Track Changes’ were highlighted.

The language of this manuscript was carefully revised by ‘AJE Language Editing’ again.

We appreciate very much for your time spent on our manuscript and the referees for their valuable suggestions and comments. The revised manuscript has been resubmitted to your journal. We are looking forward to hearing from your final decision when it is made. 

Best wishes

Sincerely, 

Yongde Yue

15-Sep-2019 

Response to reviewers:

Reviewer #3: Your manuscript submission is interesting and contains information that our readers would find useful. AND also this revision looks more better than before, Unfortunately, the quality of the language is not up to the standards of the journal. language error still can be see in the current version.

I suggest you may wish to consider having your paper professionally edited for English language by a native English speaker and/or a professional language editing service here before resubmitting this manuscript： 

http://www.zn16289208.icoc.bz/

http://www.unlecture.com/

http://www.goodenglish.com/

http://www.journalexpert.com/

which specialise in editing scientific and technical research papers written in English by authors who are not native English speakers.

Please note that while this service will greatly improve the readability of your paper, it does not guarantee acceptance of your paper by the journal.

Thanks to reviewer’s suggestion, it’s very carefully.

Response: Thank you for such high rating to the results of this manuscript, and we are regret to the quality of the language. Then, we have polished and corrected typos and mistakes, and the language of this manuscript was carefully revised by ‘AJE Language Editing’ according your suggestion. 

Thanks to reviewer’s comments and suggestions.

---

## [Editor Report · Decision Letter 2]

24 Sep 2019

PONE-D-19-17188R2

Photo-degradation dynamics of five neonicotinoids: bamboo vinegar as a synergistic agent for improved functional duration

PLOS ONE

Dear Dr. Yue,

Thank you for submitting your manuscript to PLOS ONE. After careful consideration, we feel that it has merit but does not fully meet PLOS ONE’s publication criteria as it currently stands. Therefore, we invite you to submit a revised version of the manuscript that addresses the points raised during the review process.

ACADEMIC EDITOR: although the  revised ms looks much better,still lots of language mistakes can be easy found, again, this is the last time to reminders.

We would appreciate receiving your revised manuscript by Nov 08 2019 11:59PM. To enhance the reproducibility of your results, we recommend that if applicable you deposit your laboratory protocols in protocols.io, where a protocol can be assigned its own identifier (DOI) such that it can be cited independently in the future. For instructions see: http://journals.plos.org/plosone/s/submission-guidelines#loc-laboratory-protocols

We look forward to receiving your revised manuscript.

Kind regards,

Yulin Gao

Academic Editor

PLOS ONE

---

## [Author Response · Author response to Decision Letter 2]

25 Sep 2019

Dear Dr Yulin Gao, 

We have revised the manuscript in response to the editor’ advice, and the following shows the point by point response to the editor’ comments. Additionally, we made some other revisions. All the changes in separate file and labeled ‘Revised Manuscript with Track Changes’ were highlighted.

The language of this manuscript was carefully revised and modified by Prof. Rafael Luque.

We appreciate very much for your time spent on our manuscript and the valuable suggestions and comments. The revised manuscript has been resubmitted to your journal. We are looking forward to hearing from your final decision when it is made. 

Best wishes

Sincerely, 

Yongde Yue

25-Sep-2019 

Response to editor:

Editor: After careful consideration, we feel that it has merit but does not fully meet PLOS ONE’s publication criteria as it currently stands. Therefore, we invite you to submit a revised version of the manuscript that addresses the points raised during the review process.

Response: Thank you for such high rating to the results of this manuscript, and we are regret to the quality of the language. Then, we have polished and corrected typos and mistakes, and the language of this manuscript was carefully revised and modified by Prof. Rafael Luque. 

Thanks to editor’s comments and suggestions.

---

## [Editor Report · Decision Letter 3]

27 Sep 2019

Photo-degradation dynamics of five neonicotinoids: bamboo vinegar as a synergistic agent for improved functional duration

PONE-D-19-17188R3

Dear Dr. Yue,

We are pleased to inform you that your manuscript has been judged scientifically suitable for publication and will be formally accepted for publication once it complies with all outstanding technical requirements.

With kind regards,

Yulin Gao

Academic Editor

PLOS ONE

---

## [Editor Report · Acceptance letter]

7 Oct 2019

PONE-D-19-17188R3 

Photo-degradation dynamics of five neonicotinoids: bamboo vinegar as a synergistic agent for improved functional duration 

Dear Dr. Yue:

I am pleased to inform you that your manuscript has been deemed suitable for publication in PLOS ONE. Congratulations! Your manuscript is now with our production department. 

With kind regards,

on behalf of

Dr. Yulin Gao 

Academic Editor

PLOS ONE